# VT68.2: An Antibody to Chondroitin Sulfate Proteoglycan 4 (CSPG4) Displays Reactivity against a Tumor-Associated Carbohydrate Antigen

**DOI:** 10.3390/ijms24032506

**Published:** 2023-01-28

**Authors:** Bernice Nounamo, Fariba Jousheghany, Eric Robb Siegel, Steven R. Post, Thomas Kelly, Soldano Ferrone, Thomas Kieber-Emmons, Behjatolah Monzavi-Karbassi

**Affiliations:** 1Department of Pathology, University of Arkansas for Medical Sciences, 4301 W. Markham St., Little Rock, AR 72205, USA; 2Department of Biostatistics, University of Arkansas for Medical Sciences, 4301 W. Markham St., Little Rock, AR 72205, USA; 3Winthrop P. Rockefeller Cancer Institute, University of Arkansas for Medical Sciences, 4301 W. Markham St., Little Rock, AR 72205, USA; 4Department of Surgery, Massachusetts General Hospital, Yaw 7, 55 Fruit St, Boston, MA 02114, USA

**Keywords:** CSPG4, TACAs, breast cancer, carbohydrate-mimicking peptide

## Abstract

The anti-CSPG4 monoclonal antibodies (mAbs) have shown anti-tumor activity and therapeutic potential for treating breast cancer. In addition, CSPG4 is a dominant tumor-associated antigen that is also involved in normal-tissue development in humans. Therefore, the potential for off-tumor activity remains a serious concern when targeting CSPG4 therapeutically. Previous work suggested that glycans contribute to the binding of specific anti-CSPG4 antibodies to tumor cells, but the specificity and importance of this contribution are unknown. In this study, the reactivity of anti-CSPG4 mAbs was characterized with a peptide mimetic of carbohydrate antigens expressed in breast cancer. ELISA, flow cytometry, and microarray assays were used to screen mAbs for their ability to bind to carbohydrate-mimicking peptides (CMPs), cancer cells, and glycans. The mAb VT68.2 displayed a distinctly strong binding to a CMP (P10s) and bound to triple-negative breast cancer cells. In addition, VT68.2 showed a higher affinity for N-linked glycans that contain terminal fucose and fucosylated lactosamines. The functional assays demonstrated that VT68.2 inhibited cancer cell migration. These results define the glycoform reactivity of an anti-CSPG4 antibody and may lead to the development of less toxic therapeutic approaches that target tumor-specific glyco-peptides.

## 1. Introduction

The aberrant glycosylation results in the cell-surface expression of tumor-associated carbohydrate antigens (TACAs) and is a hallmark of carcinogenesis [1,2]. As a result, there is a substantial interest in developing tumor-reactive antibodies with protein-glycan recognition properties for cancer therapy [3,4]. For example, 5E5, a MUC1-Tn/STn specific antibody that recognizes both peptide and glycan structures, shows remarkable cancer specificity and is in clinical testing [3]. In addition, Canals Hernaez et al. reported the cancer specificity of a mAb to podocalyxin, POD0447, that specifically binds a tumor-restricted glycoform of podocalyxin but has no reactivity towards podocalyxin expressed by normal adult tissue [4]. The results of a subsequent study indicate that the tumor specificity of POD0447 results from its ability to bind to the Thomsen-Friedenreich (TF) antigen (Galβ1-3GalNAcα1-Ser/Thr), a well-studied TACA [5] expressed on a podocalyxin backbone.

The chondroitin sulfate proteoglycan 4 (CSPG4) is an immunodominant tumor-associated antigen expressed in a variety of cancers, including melanoma, glioma, mesothelioma, neuroblastoma, adult and pediatric sarcomas, triple-negative breast cancer (TNBC), squamous-cell carcinoma, oligodendrocytoma, and some hematologic cancers [6,7,8,9,10]. CSPG4 was investigated as a potential immunotherapy target due to its restricted expression in normal tissues, increased expression in tumors at different disease stages, and ability to support tumor growth and progression [9]. The CSPG4 contributes to malignancy and disease progression by promoting cancer cell adhesion, motility, and survival [8,11,12,13]. In addition, anti-CSPG4 mAbs inhibit the proliferation, adhesion, and migration of TNBC [14,15]. The targeting of CSPG4 with mAbs in preclinical experiments showed potential anti-tumor activity with no apparent harmful side effects [11]. However, CSPG4 expression changes during the development of several normal tissues, suggesting that it plays a role in the formation of adult tissues [16]. Therefore, concerns related to the tumor specificity and on-target/off-tumor activity of therapeutic approaches hamper the development of clinical strategies using anti-CSPG4 mAbs. 

Additionally, Drake et al. [10] reported the possible involvement of carbohydrate moieties in the binding of anti-CPG4 mAbs to tumor cells. Therefore, the current study was performed to define the glycan specificity of anti-CSPG4 antibodies and to understand the glycan dependency of CSPG4 reactivity with triple-negative breast cancer (TNBC) cells. Our results show that the mAb VT68.2 binds to the carbohydrate-mimicking peptide 10s (P10s) and to N-linked glycans on breast cancer cells. We further show that VT68.2 is reactive with fucosylated glycans and has similar anti-tumor functionality as anti-P10s serum antibodies.

## 2. Results

### 2.1. VT68.2 Binding to MDA-MB-231 Cells Involves Glycans

The potential contribution of glycans to the binding of anti-CSPG4 mAbs 225.28 and VT68.2 to leukemia cells was reported [10]. We demonstrated that CSPG4 is expressed on triple-negative MDA-MB-231 but not on luminal ER+ MCF7 cells [17]. Therefore, we examined the binding of 225.28 and VT68.2 antibodies to MDA-MB-231 cells and showed that both antibodies bound to these cells (Figure 1A,B). In addition, we tested the reactivity of VT68.2 against additional tumor cell types and found that the mAb has reactivity towards melanoma and lung cell lines (Appendix A).

In order to determine if glycans contribute to mAb binding to breast cancer cells, MDA-MB-231 cells were treated with N- or O-linked glycosidases to remove specific glycans before examining antibody binding. Treating cells with O-glycosidase did not affect the subsequent binding of either antibody to the cells (Figure 1A,C). However, the binding of VT68.2, but not 225.28, to cells was inhibited by N-glycosidase treatment (Figure 1A,C), indicating that N-linked glycans contribute to VT68.2 reactivity with CSPG4 on MDA-MB-231 breast cancer cells. 

We used a commercially available glycan microarray to define the glycan binding specificity of VT68.2. Fucose, glucose, galactose, and fucosylated lactosamines, including the Lewis A (LeA), sialylated LeA, and Lewis Y (LeY) carbohydrate antigens, were among the top ten VT68.2 reactive sugars (Table 1). Further analysis of the array results using MotifFinder [18] indicated that β-glucose and α-fucose are the primary binding motifs recognized by VT68.2 (Table 2). β-glucose is represented on the glycan array by the monosaccharide glucose and natural glucose oligomers such as cellulose. In contrast, α-fucose is a cancer-relevant moiety for O- and N-linked TACAs. α-Fucose was the primary binding motif identified in 19 glycan structures recognized by VT68.2 (Table 2). These data are consistent with VT68.2 binding to α-fucose and α-fucose-containing tumor-associated glycans. 

### 2.2. Anti-CSPG4 VT68.2 mAb Binds to P10s

The CMP P10s structurally mimic a moiety found in multiple glycan antigens, including fucosylated lactosamines LeY and GD2 ganglioside. Immunization with PADRE-conjugated P10s induced the production of anti-LeY and anti-GD2 antibodies in humans [19]. Due to the fact that P10s is a functional mimotope capable of mimicking several glycans, we examined the reactivity of VT68.2 and an array of other anti-CSPG4 mAb clones against P10s by ELISA. Notably, among the antibodies tested, we observed that VT68.2 bound strongly to immobilized P10s (Figure 2A,B). The lack of homology between the peptide sequence and VT68.2′s target antigen, CSPG4 (Figure 2C), suggests that the ability of VT68.2, but not other anti-CSPG4 mAbs, to bind P10s is due to the anti-glycan reactivity of this mAb. 

### 2.3. Anti-P10s Serum Antibodies Share Specificity with VT68.2

The results above suggest that VT68.2 might interact with fucosylated glycans such as LeY. Therefore, we compared the ability of VT68.2 and a well-characterized anti-LeY mAb, BR55-2, to bind P10s and a CMP, P104, that specifically mimics the LeY antigen [20]. VT68.2 and BR55-2 bound to P104 and P10s (Figure 3A,B), although VT68.2 showed greater binding to P10s. Interestingly, the VT68.2 mAb inhibited the binding of IgG serum antibodies from patients immunized with PADRE-conjugated P10s to immobilized P10s (Figure 3C). These results indicate structural similarity between the target of VT68.2, the P10s mimotope, and the LeY antigen.

### 2.4. VT68.2 Inhibits Migration of Tumor Cells

We previously reported that serum antibodies from patients vaccinated with PADRE-conjugated P10s induced cytotoxic and anti-migratory effects on tumor cells [21]. The possibility that VT68.2 would elicit similar effects is suggested by its reactivity with P10s and similar TACAs. To test this possibility, we treated MDA-MB-231 cells with VT68.2 or a control antibody for up to 96 h. In contrast to the described cytotoxic effects of P10s anti-serum, we did not detect any signs of cell toxicity or growth cycle arrest (Figure 4A and Appendix A). However, using a Boyden chamber assay, we showed that VT68.2 significantly inhibited the migration of MDA-MB-231 cells (Figure 4B). 

## 3. Discussion

Given its role in malignancy and its immunogenicity, CSPG4 is a promising target for developing therapeutic approaches for solid tumors, including triple-negative breast cancer [22,23]. However, anti-CSPG4 mAbs have shown promise as active anti-cancer reagents [20,22,23,24,25]. Moreover, strategies using mAbs against CSPG4 could be effective against both primary breast tumors and their metastases [22]. The CSPG4 is also expressed in cancer-associated fibroblasts and a subpopulation of stromal cells with a role in tumor invasion and growth [26]. Therefore, using CSPG4 as a CAR-T target in solid tumors is attractive because doing so may give CAR-T cells the ability to attack primary and metastatic tumor cells and the tumor-supportive stroma [22]. However, the potential for off-tumor/on-target activity, such as against pericytes, is considered a serious caveat for clinical testing of aggressive anti-CSPG4-dependent approaches [22]. In contrast, antibodies directed toward TACAs present on CSPG4 may have greater cancer specificity, thereby providing additional protection for normal tissues. Our data indicate that VT68.2 binds to CSPG4-positive triple-negative breast cancer cells and that N-glycosidase treatment reduces this binding. Others showed that the binding of this mAb was inhibited when cells were treated with tunicamycin [10]. The tunicamycin treatment blocks N-glycosylation and has been historically used to show glycan dependency in various settings [27,28,29,30].

VT68.2 strongly binds to P10s, a CMP that mimics both LeY and GD2. We observed that anti-P10s serum antibodies reacted with both GD2 and LeY and inhibited cell migration and induced cell death [19,21]. However, VT68.2 inhibits tumor-cell migration but does not cause direct toxicity. This is consistent with CSPG4′s role in the motility and migration of tumor cells [14]. Our results indicate that the fucosylated glycans may contribute to VT68.2 binding to CSPG4 on MDA-MB-231 cells. However, we must emphasize that the data characterizing VT68.2′s glycan specificity needs cautious interpretation because the cell-surface expression of TACAs and their presentation by protein carriers significantly affect glycan reactivity. For example, glycan array analysis of the anti-podocalyxin mAb POD0447 indicated binding to N-acetylgalactosamine [4], but further analysis using a cell-based glycan array model showed that the TF antigen was the POD0447-reactive glycan [5]. Therefore, additional studies are needed to further characterize the glycoconjugates involved in VT68.2’s binding to tumor cells.

In summary, the VT68.2 monoclonal antibody reacts with several cancer cell lines, including the triple-negative breast cancer cell line MDA-MB-231. We found that VT68.2 binding to breast cancer cells involves the expression of N-glycan structures and inhibits cell migration. Although further studies are needed to confirm the specific tumor-associated glycans that mediate VT62.8 binding, our results support exploring the therapeutic use of this antibody to treat triple-negative breast disease. 

## 4. Materials and Methods

### 4.1. Reagents

The antibodies used included the following anti-CSPG4 mAb clones: VT68.2, 225.28, 763.74, TP32, VF1-TP41.2, VF1-TP43, VF4-TP108, VF4-TP109, VF20-VT5.1-3G10.1, VT80.12, and 149.53 [31,32]. A mouse anti-human CD4 (Cat# 555344, RRID:AB_395749, BD Biosciences, Franklin Lakes, NJ, USA) was used as an isotype control in flow cytometry and migration assays. For flow cytometry assays, a R-Phycoerythrin-conjugated polyclonal goat anti-mouse F(ab’)_2_ fragment from Dakocytomation (Agilent Dako, code No. R0480, Santa Clara, CA, USA) was used as a secondary antibody. Peroxidase-conjugated goat anti-mouse IgG (Cat# A3673, RRID:AB_258099, Sigma-Aldrich, Inc., Saint Louis, MO, USA) was used for binding in ELISA assays. Biotin-conjugated goat anti-mouse IgG (Sigma-Aldrich Cat# B1140, RRID:AB_258513) was used for glycan array experiments. Peroxidase-conjugated goat anti-human IgG (Cat# A6029, RRID:AB_258272, Sigma-Aldrich, Inc.) was used for the inhibition assays performed using human sera immunized with PADRE-cojugated P10s. GIBCO™ enzyme-free cell dissociation buffer was purchased from Fisher Scientific (Waltham, MA, USA). Neuraminidase, O-glycosidase, and N-glycosidase (Roche Inc., Basel, Switzerland) were used to treat cancer cells. The DIF-Quick kit (Dade Behring Inc., Deerfield, IL, USA) was used for counting migrated cells in migration assays. Other reagents included two CMPs manufactured as multiple antigenic peptides (MAP): P104 (GGIMILLIFSLLWFGGA, Research Genetics, Huntsville, AL, USA) and P10s (WRYTAPVHLGDG, AmbioPharm, Inc., North Augusta, SC, USA). 

### 4.2. Cell Lines

Human breast, melanoma, and lung cancer cell lines were from ATCC (Manassas, VA, USA). Cells were cultured in a base medium supplemented with 10% heat-inactivated fetal bovine serum (Life Technologies), 50 units/mL penicillin, and 50 μg/mL streptomycin. The base medium for MDA-MB-231 was DMEM (Thermo Fisher Scientific Inc., Waltham, MA, USA). The base medium for MCF7 was MEM (Thermo Fisher Scientific Inc.) supplemented with 0.1 mM non-essential amino acids, 1 mM sodium pyruvate, and 0.01 mg/mL insulin (Thermo Fisher Scientific Inc.). Melanoma and lung cells were cultured in RPMI-1640. 

### 4.3. Flow Cytometry

The binding of anti-CSPG4 mAbs to cells was determined using flow cytometry, as previously described [33]. Briefly, cells were grown for 48 h and then the subconfluent cells were harvested using GIBCO™ enzyme-free cell dissociation buffer and then were washed and resuspended in FACS buffer (PBS with 1% BSA). The resuspended cells were incubated on ice with 5 to 10 µg/mL of anti-CSPG4 mAbs or an equal amount of the isotype control for 30 min. After washing two times, R-phycoerythrin-conjugated polyclonal goat anti-mouse F(ab’)_2_ was added in a 1:500 dilution as the secondary antibody to detect anti-CSPG4 binding. Samples were washed twice and acquired on BD Biosciences LSRFortessa. Data were analyzed using FlowJo (BD, Franklin Lakes, NJ, USA). 

### 4.4. Glycosidase Treatment

In order to evaluate a role for glycans in VT68.2 binding to MDA-MB-231 cells, 1.5 × 10^5^ cells were incubated in a serum-free medium containing N-Glycosidase (5 U/mL, Roche Inc.) for 3 h at 37 °C. When treating cells with O-glycosidase, cells were first incubated with neuraminidase (100 mU/mL, Roche Inc.) for 1 h at 37 °C; O-glycosidase (10 mU/mL, Roche Inc.) was then added, and the mix was incubated at 37 °C for an additional 2 h. Following treatment with glycosidases, cells were washed with FACS buffer (PBS with 1% BSA) and then stained with VT68.2 mAb as described above.

### 4.5. Cytotoxicity Assay

Additionally, to assess the cytotoxicity of the mAb towards the cell line, crystal violet staining of attached cells was performed [34]. Further, 5 × 10^4^ of the respective cells were seeded into the wells of a 24-well plate to grow in medium containing 10% FBS. After 24 h, the wells were refreshed with medium containing 20 µg/mL VT68.2, which was refreshed every 24 h by replacing half the medium. At various intervals (24, 48, 72, and 96 h) after the first addition of the mAb, supernatants were removed from replicated wells, wells were washed with PBS, and live cells were stained with Crystal Violet. After final staining at day 5, the wells were photographed, the stain was solubilized with adding 1% SDS, and the plate was read by a Synergy LX multi-mode reader (BioTek, Winooski, VT, USA) at 570 nm to assess cell number. 

### 4.6. Migration Assay

The cell migration was assayed using Corning^®^ Transwell Permeable Supports (Corning, Inc., Corning, NY, USA) with polycarbonate membranes (6.5 mm, 8-μm pore size). Cells were grown for 48 h and then starved overnight. Cells were lifted using GIBCO™ enzyme-free cell dissociation buffer, washed, and then incubated with 5 µg/mL of the isotype control or VT68.2 mAb for 30 min at room temperature. The mixture was transferred into the upper chambers of the transwell plates for incubation. The lower chamber of each well was filled with 600 µL of complete medium (10% FBS). After 48 h of incubation, the cells were fixed and stained using a DIF-Quick kit (Dade Behring Inc.). Non-migrating cells on the top side of the membrane were removed by gently scrubbing with cotton swabs. Migrated cells were counted in five randomly chosen fields under a light microscope.

### 4.7. ELISA Assay

ELISA was used to assess anti-CSPG4 mAbs’ binding to CMPs and to inhibit anti-P10s serum antibodies’ binding back to the P10s peptide by VT68.2 mAb. For binding assays, ELISA plates (Immuno 4 HBX, Thermo Fisher Scientific) were coated overnight with 50 µL per well of 10ug/mL of peptides. After blocking (blocking buffer: PBS with 1% BSA), serially diluted antibodies were added to wells in blocking buffer and incubated at 37 °C for 2 h. Following washing at room temperature, goat anti-mouse IgG secondary antibody (Sigma-Aldrich, Inc., Saint Louis, MO, USA) was added in 1:15,000 dilution in blocking buffer for an hour’s incubation at 37 °C. The inhibition assay was performed similarly, except PADRE-conjugated P10s-immunized human serum [19] was added in a 1:800 dilution after washing off VT68.2 antibody and binding of the immunized serum antibodies to the peptide were visualized with an HRP-conjugated anti-human IgG (Sigma-Aldrich, Inc.), at a dilution of 1:10,000. Binding was detected by measuring absorbance at 450 nm using the Synergy LX multi-mode reader (BioTek©, Winooski, VT, USA). 

### 4.8. Glycan Array

The Glycan 100 Microarray Kit (RayBiotech Life, Inc., Peachtree Corners, GA, USA) was used to determine the glycan motif and binding specificities of the VT62.8 mAb. The experiments followed the manufacturer’s guidelines. Further, slides were incubated overnight with 10 µg/Ml of VT68.2 in sample diluent buffer, washed once with Wash Buffer I and twice with Wash Buffer II for 5 min each at RT with gentle shaking. Then the slide was incubated with the biotinylated anti-mouse IgG (Sigma-Aldrich, Inc.) in a 1:15,000 dilution for 2 h. After washing with the Wash Buffers two times each as described above, binding was visualized with the streptavidin-Cy3 conjugate provided by the kit, followed by extensive washings as instructed by the manufacturer. A fluorescence microarray scanner (Innopsys, Carbonne, France) and MAPIX software were used to acquire the data. Background-subtracted median fluorescence intensities were generated. Additionally, we used MotifFinder software version 3.1.0 with its default settings to build an automated reactivity model and deduce the glycan binding of the VT68.2 antibody to a common glycan motif [18,35]. Raw data were processed to remove antibiotic spots and glycans with undefined structures to make files compatible with the MotifFinder glycan library (Appendix A). This file was also used to sort glycans with the highest binding capacity. MotifFinder was run on 6 January 2023. 

### 4.9. Statistical Analyses

Assays were repeated independently at least twice, and the means and SDs/SEMs were calculated. Paired *t*-test, 1-sample *t*-test and randomized-block analysis of variance (ANOVA) with Dunnett’s multiple comparisons test were used to compare the means at a significance level of α = 0.05. *p* values were two-tailed. Prism 9 (GraphPad Software, Inc., San Diego, CA, USA) software was used for statistical analyses. 

## Figures and Tables

**Figure 1 ijms-24-02506-f001:**
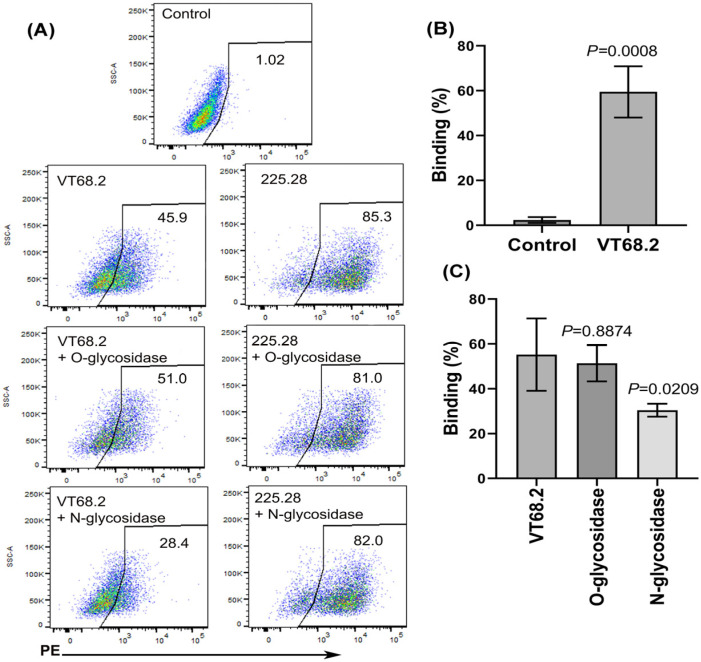
Glycans are involved in VT68.2 binding to MDA-MB-231 cells. Anti-CSPG4 mAbs 225.28 and VT68.2 bind to MDA-MB-231 (**A**, top panel). Cells were stained with 225.28 or VT68.2 mAbs, and binding was visualized with an RPE-labeled polyclonal goat anti-mouse F’(ab’)2 antibody. Relative to the control treatment, treating cells with N-glycosidase (**A**, bottom panel) but not O-glycosidase (middle panel) reduced VT68.2 but not 225.8 binding to MDA-MB-231 cells. Assays of VT68.2 binding to untreated (**B**) and glycosidase-treated MDA-MB-231 cells (**C**) were repeated 3 to 4 times. Paired *t*-test (**B**) or randomized-block ANOVA (**C**) were used for statistical comparisons. Means ± SDs are shown.

**Figure 2 ijms-24-02506-f002:**
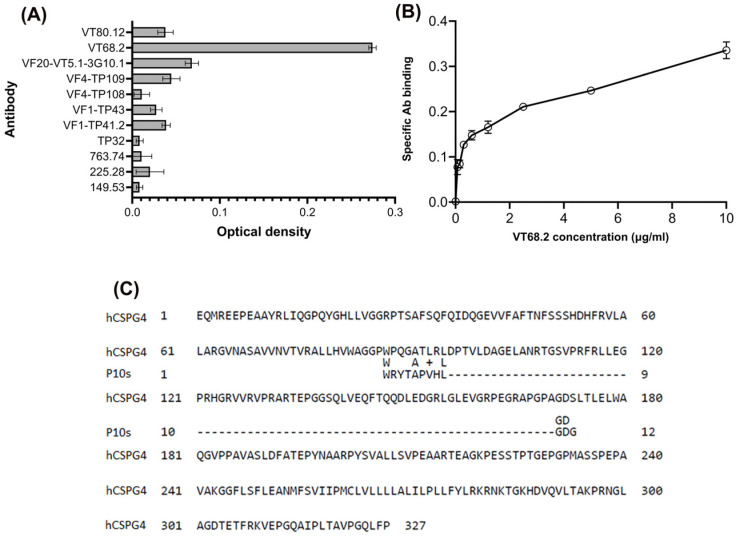
Binding of anti-CSPG4 mAbs to P10s. Anti-CSPG4 mAb VT68.2 showed stronger binding to the P10s than other mAbs. ELISA plates were coated with 10 µg/mL of P10s overnight. After washing, coated wells were incubated with serial dilutions of the indicated anti-CSPG4 mAb clones. (**A**) Reactivity of anti-CSPG4 mAbs (5 µg/mL) with P10s, showing distinct binding of VT68.2. Base-line binding with no Ab was subtracted. (**B**) Binding of VT68.2 to P10s after subtracting the antibody binding to uncoated wssells in all dilutions shows specific VT68.2 binding to P10s. The assay was independently repeated twice. Means ± SEMs are shown. (**C**) Screen display of sequence alignment of human protein CSPG4 and the peptide sequence of P10s, showing their lack of homology, using the National Center for Biotechnology Information (NCBI) BLAST.

**Figure 3 ijms-24-02506-f003:**
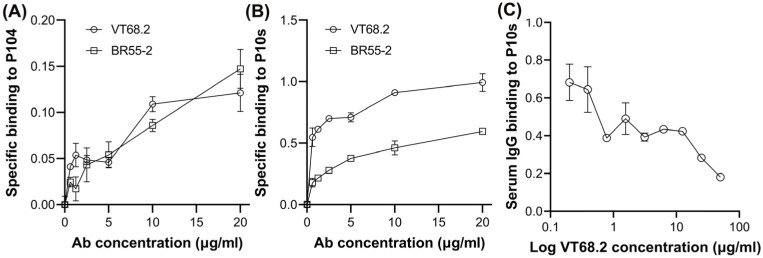
VT68.2 mAb binds to LeY-mimicking peptides and inhibits the binding of anti-P10s serum antibodies to P10s. VT68.2 and anti-Lewis Y mAb BR55-2 react with P104 (**A**) and P10s (**B**). ELISA plates were coated overnight with the peptides, and reactivity of serial dilutions of the mAbs was detected. Background binding was subtracted. (**C**) ELISA plates were coated with P10s overnight, as described above. After blocking, plates were incubated with serial dilutions of VT68.2 mAb. Plates were then washed and incubated with post-immune serum from a patient immunized with conjugated P10s. The binding of IgG in patient serum was detected using an HRP-conjugated anti-human IgG antibody. Means ± SDs are shown.

**Figure 4 ijms-24-02506-f004:**
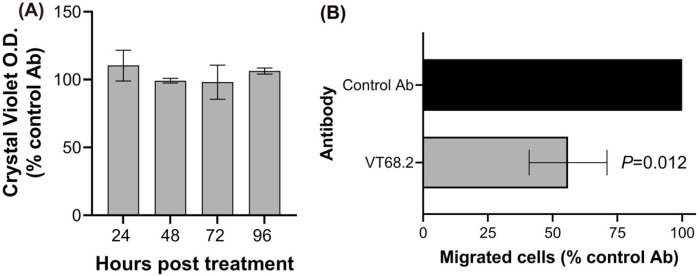
VT68.2 mAb did not induce direct cytotoxicity but inhibited the migration of MDA-MB-231 cells and other cancer cell lines. (**A**) Cells were seeded at day 0 in the presence of 5 µg/mL of the isotype control or VT68.2 mAb. Cells were harvested after 24, 48, 72, and 96 h and then stained with crystal violet. The stain was then solubilized, and the absorbance was measured at 570 nm. The ODs of wells treated with VT68.2 mAb were normalized to those of a control mAb. Means ± SD of normalized percent O.D. of VT68.2 treated cells for all time points are shown. One-sample *t*-test comparing each time point with 100% on log-transformed data showed no significant differences. (**B**) Cells were starved overnight in a serum-free medium, harvested using an enzyme-free buffer, then incubated with the isotype control, or VT68.2, transferred to transwell plates, and incubated at 37 °C for 48 h. The non-migrated cells on the top side of the membrane were wiped out, and migrated cells were counted under a light microscope. The figure portrays the average ± SEM of the number of migrated cells in the presence of VT68.2 after normalizing to isotype controls in each of 10 independent experiments. The *p* value was estimated by applying the paired *t*-test to log-transformed data.

**Table 1 ijms-24-02506-t001:** The top glycan array binders. VT68.2 was used at 10 µg/mL to react with 100-glycan microarray.

Glycan	MFI
α-Fuc-Sp	3422.5
Glcβ1-4Glcβ1-4Glcβ1-4Glcβ-Sp1	962.75
β-Glc-Sp	658.13
α-Man-Sp	507.5
β-Gal-Sp	342.38
Neu5Acα-2,3Galβ-1,3(Fucα-1,4)GlcNAcβ-Sp (Sialyl Lewis A)	319.38
(Fucα1-2)Galβ1-4(Fucα1-3)GlcNAcβ-Sp1 (Lewis Y)	299.63
Galβ1-3(Fucα1-4)GlcNAcβ-Sp (Lewis A)	249.50
GlcNAcβ1-6GlcNAcβ-Sp	238.00
Glcα1-2Galα1-3Glcα-Sp	237.00
Negative control + 3(SD)	7.16

MFI, Median Fluorescence Intensity after background subtraction (range −1.63 to 3348.25). Glycans were sorted based on MFI with the top 10 glycan binders shown

**Table 2 ijms-24-02506-t002:** MotifFinder Identified Primary Binding Motifs. Results from the glycan array binding experiment using VT68.2 were analyzed by MotifFinder to identify primary terminal VT68.2-binding motifs.

Motif Name	Graphic Structure	No. of Glycans	Reactive Glycans
α-Fucose	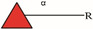	19	α-Fucose and fucosylated lactosamines
β-Glucose	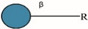	5	β-Glucose and cellulose oligomers
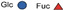

## Data Availability

The data presented in this study are available in the article or as Appendix A.

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
