# Peer review of "VT68.2: An Antibody to Chondroitin Sulfate Proteoglycan 4 (CSPG4) Displays Reactivity against a Tumor-Associated Carbohydrate Antigen"

_ijms, 2023, doi:10.3390/ijms24032506_

Round 1
Reviewer 1 Report
In manuscript IJMS-2045896, Nounamo and colleagues have demonstrated mAb VT68.2 binds to N-linked glycans and carbohydrate mimicking peptide 10s on breast cancer cell MDA-MB-231. Further assay showed VT68.2 inhibited MDA-MB-231 cell migration. This study has a compact story yet the methodology and experimental results are not strong enough to support the conclusion. Another concern is that the reproducibility of the data from this study. Here are the specific Comments:
(1) The whole study mainly focus only in one cell line MDA-MB-231. Since CSPG4 was overexpressed in many cancer cells such as melanoma cells it would be more plausible to test VT68.2 in some of the other cancer cell lines for all the panels.
(2) Table 2 was not intact because the bottom row listed six symbols. Better to show the complete list of reactive glycan.
(3) For FACS panels in Figure 1 and Figure 4, authors should also show multiple repeats with statistical analyses.
(4) Figure 3 was confusing not only because of the labeling (no unit label in A and B, concentration gradient was opposite in A, B and C) but also the results of in B and C was not consistent.
(5) For Figure 4A, the methodology for proliferation assay was not good enough to meet the current publication standard.
(6) For Figure 4B, the migration assay showed very large variance from 100 range to 1000 range which raised questions on the confidence of the experiments.
(7) For Figure 4C, the labels of PE was poorly aligned.
(8) Line 68, should be MDA-MB-231.
(9) Line 138, two periods at the end.
Reviewer 2 Report
Nounamo and coworkers present a manuscript characterizing the glycan binding of an anti-CPG4 monoclonal antibody towards established breast cancer cell lines and a 100 glycan array. They demonstrate that functionally, the antibody does not inhibit cancer cell growth but reduces migration of cancer cells in an in vitro transwell assay. The results provide preliminary proof-of-concept data to study these effects in an in vivo mouse model of the human cancer cell lines for therapeutic activity and provide a therapeutic target for triple-negative breast cancers. A thorough review of the manuscript for English grammar is recommended.
Minor Concerns:
1) Line 75 – Treatment of cells with glycosidases. There is no description of this procedure in the materials and methods section if results needed to be replicated.
2) Figure 1 – X-axis label missing in plots
3) Table 1 legend – “VT68.2 was used at 10 and µg/ml to react…” suspected typo and should be corrected
4) Figure 2 – Does the ELISA signal for the VT68.2 antibody return to baseline with increased dilution? It is not clear from this data if the antibody may bind non-specifically to the plate and cause an increased baseline for that particular antibody.
5) Figure 3 – What are the units for the Ab concentration on the x-axis for this plot? Some methods were included in the figure legend that might be better placed in the materials and methods section. What concentration of peptides was used to coat the plates overnight?
6) Line 156 – Several locations in the manuscript reference “570nM” where the authors intend to measure the absorbance at 570 nm.
7) Line 229 – What concentration of antibodies were used for the flow cytometry experiments. Additional information is needed on how cells were analyzed on the cytometer, what cytometer was used, etc.
8) Line 245 – What is the composition of the enzyme-free buffer?
9) Line 250 – Who is the manufacturer of the DIF-Quick kit?
10) Line 255 – What was the composition of the blocking buffer and what concentration was used for blocking.
11) Line 257 – What concentration/dilution of secondary antibody was used for the ELISA experiments. Is there an RRID number provided for the particular antibodies used in this study?
Round 2
Reviewer 1 Report
The authors addressed all the concerns, I agree with the acceptance of this paper for publication.